# Enforced Mutualism Leads to Improved Cooperative Behavior between *Saccharomyces cerevisiae* and *Lactobacillus plantarum*

**DOI:** 10.3390/microorganisms8081109

**Published:** 2020-07-24

**Authors:** S. Christine du Toit, Debra Rossouw, Maret du Toit, Florian F. Bauer

**Affiliations:** South African Grape and Wine Research Institute, Department of Viticulture and Oenology, University of Stellenbosch, Private Bag X1, Matieland, Stellenbosch 7602, South Africa; scdt@sun.ac.za (S.C.d.T.); debra@sun.ac.za (D.R.); mdt@sun.ac.za (M.d.T.)

**Keywords:** *Saccharomyces cerevisiae*, *Lactobacillus plantarum*, synthetic ecology, co-evolution

## Abstract

*Saccharomyces cerevisiae* and *Lactobacillus plantarum* are responsible for alcoholic and malolactic fermentation, respectively. Successful completion of both fermentations is essential for many styles of wine, and an understanding of how these species interact with each other, as well as the development of compatible pairings of these species, will help to manage the process. However, targeted improvements of species interactions are difficult to perform, in part because of the chemical and biological complexity of natural grape juice. Synthetic ecological systems reduce this complexity and can overcome these difficulties. In such synthetic systems, mutualistic growth of different species can be enforced through the reciprocal exchange of essential nutrients. Here, we implemented a novel approach to evolve mutualistic traits by establishing a co-dependent relationship between *S. cerevisiae* BY4742Δ*thi4* and *Lb. plantarum* IWBT B038 by omitting different combinations of amino acids from a chemically defined synthetic medium simulating standard grape juice. After optimization, the two species were able to support the growth of each other when grown in the absence of appropriate combinations of amino acids. In these obligatory mutualistic conditions, BY4742Δ*thi4* and IWBT B038 were co-evolved for approximately 100 generations. The selected evolved isolates showed improved mutualistic growth and the growth patterns under non-selective conditions indicate the emergence of mutually beneficial adaptations independent of the synthetic selection pressure. The combined use of synthetic ecology and co-evolution is a promising strategy to better understand and biotechnologically improve microbial interactions.

## 1. Introduction

Grape juice contains a complex, interactive microbial community of various yeast and bacteria species responsible for the biochemical conversion of must to wine. Initially, various non-*Saccharomyces* yeast species are present, some of which may contribute to the organoleptic properties of the wine, but *Saccharomyces cerevisiae* outcompetes these species to dominate the later stages of the fermentation [1,2]. *S. cerevisiae* is therefore the primary yeast species responsible for the completion of alcoholic fermentation (AF), where the sugars glucose and fructose in the grape must are converted to ethanol. During AF, the ethanol concentration increases and many nutrients are utilized by the yeast. This creates a harsh environment for lactic acid bacteria (LAB) to subsequently grow in and complete the secondary fermentation, where malic acid is converted to lactic acid, known as malolactic fermentation (MLF) [3]. *Oenococcus oeni* is the primary species used in commercial starter cultures to initiate MLF, however the focus has shifted to other LAB species such as *Lactobacillus plantarum* as possible alternatives [4].

Simultaneous inoculation of yeast and LAB can possibly overcome some of these difficulties, since LAB are expected to adapt better to environmental conditions in grape must rather than wine [5]. However, this practice results in the yeast and bacteria forming biomass and being metabolically active while in direct contact with each other. Substantial research has been done on how yeast and bacteria strains interact under winemaking conditions, but there is still much that is not known (for a review, see [6]). Nutrient competition is one factor that could inhibit the growth of these organisms. LAB have limited biosynthetic capabilities and are auxotrophic for various amino acids and vitamins [4,7]. In co-culture, MLF may be inhibited if the yeast rapidly depletes these nutrients or AF may be sluggish if the LAB consumes all available trace elements and other survival factors [2,8]. Competition for biochemical compounds between yeast and LAB is, however, not yet fully understood [2].

Numerous factors affect the growth of *S. cerevisiae* and LAB during fermentation, including pH, temperature, and nitrogen availability, among other factors. Additionally, some interactions between the species, such as physical contact between cells and release of extracellular compounds, have also been reported [2,9,10]. Such interactions can be stimulatory or inhibitory to the growth of the organisms and the successful completion of AF and MLF, respectively. These are partly dependent on specific strains and are not always applicable at the species level [11,12]. 

Yeast species can inhibit bacterial growth and MLF through the release of medium-chain fatty acids (MCFAs) into the extracellular environment [9]. MCFAs deprotonate within cells, resulting in intracellular acidification, loss of the transmembrane gradient, and subsequent inhibition of ATPase, which is an important enzyme for MLF [13]. This inhibition is increased under low pH conditions and in the presence of ethanol, and is also dependent on the concentrations of individual MCFAs [14]. *Lactobacillus* species have been reported to inhibit yeast growth and AF through various mechanisms, such as the production of acetic acid and other short-chain carboxylic acids, which cause acidification of the yeast intracellular environment and subsequent cell death [8]. LAB also have extracellular β-1,3-glucanase activity, which means they could potentially degrade yeast cell walls, and have been reported to produce bacteriocins capable of potentially inhibiting the growth of other LAB [2,15,16].

Selecting compatible *S. cerevisiae* and LAB strains is, therefore, important for successful AF and MLF [17]. Furthermore, directed evolution has been suggested as a strategy to further improve strain compatibility and to optimize strains by increasing their resistance to inhibitory biotic and abiotic factors encountered under winemaking conditions [18,19]. The chemical and biological complexity of grape must make it difficult to study the interactions between specific strains and to monitor the evolution of these interactions over time. The use of simplified ecological systems is, therefore, necessary in order to overcome these practical limitations. 

Here, we assess the applicability of a novel synthetic ecology and co-evolution approach to better understand and optimize the interactions between *S. cerevisiae* and *Lb. plantarum* strains. Nutrient co-dependency between the strains was used to establish obligatory mutualistic conditions, with the aim of evaluating if mutually beneficial adaptations independent of the obligatory nutrient exchange would emerge within the population over time. The advantage of the system is that the co-dependency ensures the persistence and co-existence of both species over time, and should focus the selection pressure to favor improved mutualistic traits. The strains were grown in a chemically defined synthetic medium simulating standard grape juice under mono- and co-culture conditions. Selected amino acids were omitted from the medium in an attempt to establish a co-dependent, mutualistic relationship between the species. The relationship was also evolved over approximately 100 generations to investigate how this continuous biotic selective pressure might affect the species. Furthermore, this study served as a proof of concept and established a platform for the stable co-evolution of these species. This system provides the opportunity to monitor and investigate the genetic changes involved in the process of co-evolution in multispecies, trans-kingdom ecosystems, using wine as a relevant model.

## 2. Materials and Methods

### 2.1. Microbial Species and Culture Conditions

*Saccharomyces cerevisiae* strain BY4742Δ*thi4* (EUROSCARF) and *Lb. plantarum* strain IWBT B038, obtained from the culture collection of the South African Grape and Wine Research Institute (SAGWRI) at the University of Stellenbosch, were used for all experiments. Strain IWBT B038 was previously isolated from grape must at SAGWRI. 

Strain BY4742Δ*thi4* was grown at 30 °C in yeast peptone dextrose (YPD) broth and on YPD agar plates (Biolab diagnostics, Wadenville, South Africa) supplemented with 0.025 mg/mL chloramphenicol (Roche, Sandton, South Africa) for the inhibition of bacterial growth when appropriate. Strain IWBT B038 was grown anaerobically at 30 °C in de Man, Rogosa and Sharpe (MRS) broth (MRS broth (Biolab diagnostics) with 20% preservative free apple juice (Ceres fruit juices (Pty) Limited, Paarl, South Africa) and on MRS agar plates (MRS broth with 20 g/L Bacteriological agar (Biolab diagnostics)) with 10% preservative free tomato juice (Tiger Food Brands Limited, Sandton, South Africa) and supplemented with 0.05 mg/mL Delvocid® Instant (DSM Food Specialities, Delft, The Netherlands), which contained 50% natamycin for the inhibition of yeast growth when required. The pH of the MRS broth and MRS agar plates were adjusted to pH 5.2 using 6 M HCl and the MRS broth was filter-sterilized through a 0.22 µm syringe filter before use. IWBT B038 cultures grown on MRS agar plates were anaerobically incubated using anaerobic containers with Anaerocult A (Merck, Darmstadt, Germany), as per the manufacturer’s instructions.

Wet stock freeze cultures (40% glycerol) were prepared for strains BY4742Δ*thi4* and IWBT B038 from cultures grown in YPD and MRS broths, respectively, and were stored at −80 °C in order to pre-culture all strains from the same culture stock solutions for all experiments conducted.

### 2.2. Amino Acid Auxotrophic Screening Assay

Synthetic medium (SM), which simulates standard grape juice, and agar plates containing 2% agar were prepared in order to evaluate the growth of strains BY4742Δ*thi4* and IWBT B038 in the absence of the different amino acids. Twenty different amino acid stock solutions were prepared. One stock solution contained all the amino acids, as listed in Table 1 (positive control), while the remaining 19 stock solutions contained only 18 of the listed amino acids, with one of the amino acids omitted.

Synthetic medium (SM) base medium was prepared by adding the carbon sources, acids, salts, and ammonium sulphate to distilled water. The pH of the medium was adjusted to pH 3.5 with KOH before being autoclaved [20,21]. The amino acids, trace elements, and vitamins were filter sterilized using a 0.22-µm syringe filter and added to the SM base medium along with the anaerobic factors and 0.02 mg/L uracil. An 8% agar solution was prepared in distilled water with bacteriological agar (Biolab diagnostics). After the agar solution was autoclaved, the SM medium was added to the hot agar and the solution was poured into petri dishes to solidify. The amino acid stock solutions were used to prepare different batches of SM agar plates. The negative control SM agar plates contained no amino acids.

Single colonies of strains BY4742Δ*thi4* and IWBT B038 from YPD and MRS agar plates, respectively, were streaked out on positive and negative control SM agar plates and on the 19 single amino acid omission SM plates. After incubation at 30 °C for 96 h, the growth of strains BY4742Δ*thi4* and IWBT B038 on the single amino acid omission SM plates were compared to their respective positive and negative control SM plates. All plates were streaked out in triplicate.

### 2.3. Pre-Culture and Fermentation Conditions

Monoculture and co-culture fermentations were all performed in triplicate. The yeast and bacteria strains were pre-cultured by first inoculating a single colony of BY4742Δ*thi4* and IWBT B038 in 5 mL YPD and MRS broths, respectively. The cultures were incubated at 30 °C for 14 h. Thereafter, the cultures were used to inoculate 30 mL of YPD and MRS broth at OD_600_ 0.1 (BY4742Δ*thi4*) and 0.01 (IWBT B038), respectively. These cultures were incubated at 30 °C for 14 h in order to generate cells for inoculation that were in mid-exponential growth phase. Before inoculation, the cells were washed twice and re-suspended in sterile water. BY4742Δ*thi4* and IWBT B038 were not subjected to a starvation period before inoculation, as it was assumed that any amino acids stored within the cells during pre-culturing would be essential for initial growth to establish cooperation between the strains.

All fermentations were performed in glass flasks with fermentation caps containing 80 mL SM medium (Table 1). The SM medium was prepared as previously described, except the anaerobic factors were added to each fermentation flask after the medium had been dispensed. The strains were inoculated at OD_600_ 0.1 (±1E+06 CFUs/mL, BY4742Δ*thi4*) and 0.01 (± 4E+06 CFUs/mL, IWBT B038) in both monoculture and co-culture fermentations. The initial pH of the medium was pH 3.5 and all fermentations were incubated at 20 °C. Sampling was performed at 0, 48, 120, and 192 h, except where stated otherwise, to measure the CFUs/mL, as well as glucose, fructose, and malic acid concentrations.

#### 2.3.1. Amino Acid Selective Conditions

Five amino acid treatments were initially used in order to select one treatment for further experiments, based on the results from Section 2.2. Four of the treatments contained 17 of the 19 listed amino acids (Table 1), with lysine and one of the following amino acids omitted: isoleucine, alanine, valine, or methionine. The remaining treatment contained 14 amino acids, with the following amino acids omitted: lysine, isoleucine, alanine, valine, and methionine (Table 2). The positive and un-inoculated controls contained all 19 amino acids. Fermentations were carried out for 336 h and sampling was performed at 0, 48, 120, 192, 240, and 336 h. 

#### 2.3.2. Co-Evolution of Yeast and Bacteria

BY4742Δ*thi4* and IWBT B038 were grown in co-culture batch fermentations in the absence of lysine and isoleucine, in order to co-evolve the species. All fermentations were performed in triplicate. Fermentations were carried out for 192 h and sampled at t = 0 h and t = 192 h to monitor cell counts. After 192 h, the yeast and bacteria cells were diluted 10-fold by re-inoculation into new SM media. The change in CFUs/mL was used to determine the average number of generations obtained during the co-evolution process. 

When approximately 100 generations of co-evolution were reached, the fermentations were plated on YPD and MRS agar plates and yeast and bacteria colonies were randomly selected for initial screening. These screenings were performed in 10 mL SM media and growth was monitored by measuring the optical density of the cultures every 24 h. 

### 2.4. Glucose, Fructose, and Malic Acid Measurement

An automated analyzer (Konelab Arena 20XT, Thermo Electron Corporation, Vantaa, Finland) was used to enzymatically assay or measure the concentrations of glucose (Enzytec™ Fluid D-glucose Id-No: E5140, Roche, R-Biopharm), fructose (Enzytec™ Fluid D-fructose Id-No: E5120, Roche, R-Biopharm), and L-malate (Enzytec™ Fluid L-malate Id-No: E5280, Roche, R-Biopharm). Malolactic fermentation was considered to be completed if the concentration of L-malate was equal to or less than 0.3 g/L, and alcoholic fermentation was considered to go to dryness if the total residual sugar concentration was less than 4 g/L.

### 2.5. Statistical Analysis

Data were analyzed by performing a one-way analysis of variance (ANOVA) followed by Tukey’s honest significance test (HSD) test using XLSTAT (version 2016.05.33324, Addinsoft, Paris, France). Differences between treatments and culture conditions were considered as significant when the *p*-values were below 0.05.

## 3. Results and Discussion

*Saccharomyces cerevisiae* strain BY4742Δ*thi4* was selected due to the ease of doing genomic analysis on a lab strain and because the haploid phenotype increases the probability that genetic mutations accumulated during co-evolution would cause observable phenotypic changes. Since BY4742Δ*thi4* is a deletion mutant and auxotrophic for lysine and thiamine, it was necessary to select a partner strain that would be able to support the growth of the yeast in the absence of these nutrients. This work only focused on the reciprocal exchange of amino acids between the *S. cerevisiae* and *Lb. plantarum* strains (Figure 1), but the use of BY4742Δ*thi4* means vitamin deficiency can be used as an additional synthetic ecology tool for follow up work. *Lb. plantarum* IWBT B038 was selected as the partner strain after showing good growth in the absence of lysine during the amino acid auxotrophic screening assay (data not shown). The screening also showed that IWBT B038 is auxotrophic for isoleucine, alanine, valine, and methionine, which BY4742Δ*thi4* is able to synthesize.

### 3.1. Initial Screening for Amino-Acid-Selective Conditions

The growth of IWBT B038 was strongly supported in this synthetic relationship, while BY4742Δ*thi4* growth under these initial conditions remained unaffected (Figure 2). Omission of amino acids caused changes to the total YAN concentrations of the amino acid treatments. However, the concentrations ranged between 293.55 and 322.48 mg/L YAN, which is still adequate for *S. cerevisiae* to complete AF [21]. The data suggest that the yeast strain released a sufficient amount of amino acids to support the growth of the bacterial population, while the bacterial population could not support yeast growth. It has previously been shown that *S. cerevisiae* secretes amino acids into the environment, stimulating the growth of other organisms, such as LAB [22]. While not impacting the growth of the yeast, the presence of IWBT B038 did, however, have a significant stimulatory effect on sugar consumption by BY4742Δ*thi4* (Figure 3). This suggests that the LAB triggered an adaptive response in *S. cerevisiae.* Previous work has suggested that the rapid conversion of glucose to ethanol by *S. cerevisiae* during fermentation may be a competitive strategy to eliminate ethanol-sensitive organisms, such as bacteria, from the environment [23].

In co-culture, BY4742Δ*thi4* showed an overall preference for glucose over fructose. The amount of sugar consumed for the five amino acid treatments by day 14 ranged from 4.62 to 12.28 g/L for glucose and from 1.81 to 6.56 g/L for fructose. BY4742Δ*thi4* consumed the highest amount of sugar in the Lys-Ile (12.28 g/L glucose, 6.56 g/L fructose) and Lys-Val (11.50 g/L glucose, 4.03 g/L fructose) treatments. However, this was still significantly lower than the 39.19 g/L glucose and 12.33 g/L fructose BY4742Δ*thi4* consumed for the control treatment (Figure 3). This indicates that the poor growth and sugar consumption when lysine is omitted are due to the bacteria’s inability to optimally support yeast growth and not due to other initial factors in the environment, such as high sugar concentrations and low pH. In contrast, IWBT B038 completed MLF by day 5 for all treatments when co-cultured with the yeast (data not shown). Based on these results, the Lys-Ile treatment was selected for subsequent experiments and for the co-evolution of the strains.

### 3.2. Addition of Lysine to Fermentations

The comparatively poorer growth performance of BY4742Δ*thi4* could theoretically have been the result of IWBT B038 outcompeting the yeast for essential vitamins or trace elements, as wine-related *Lactobacillus* strains are known to be auxotrophic for various nutrients [4]. An alternative and more likely explanation could be due to IWBT B038 releasing insufficient amounts of lysine to support BY4742Δ*thi4*. In order to investigate this hypothesis, lysine was added to the Lys-Ile treatment on day 2, after the exponential growth phase of IWBT B038, to a final concentration of 0.013 g/L (the amount present in the control treatment at the start of fermentation). Lysine addition had a significant effect on the growth of BY4742Δ*thi4*, as the yeast cell counts increased to 7.11 log CFUs/mL (SD = 0.03) and were similar to the values previously observed for the control treatment when all amino acids were present, proving the hypothesis of insufficient lysine being released.

### 3.3. Influence of Inoculation Dosage 

The addition of lysine to the fermentations and the subsequent improvement in the growth of BY4742Δ*thi4* confirmed that the current bacterial population density was not able to provide sufficient lysine to the yeast population. At the initial inoculation dosage, the yeast and bacteria had similar cell counts, but the yeast biomass was likely higher due to possible differences in cell size. *Lb. delbrueckii* subsp. *bulgaricus* has been shown to grow shorter cells when co-cultured with *S. cerevisiae,* possibly as a result of chemical alterations to the fermentation media by the yeast [24]. In order to evaluate the effect the inoculation dosage has on the yeast and bacteria, BY4742Δ*thi4* was inoculated at a 10-fold lower cell density. After this adjustment, there was a significant improvement in the growth of BY4742Δ*thi4*, while the growth of IWBT B038 was still strongly supported by the smaller yeast population (Figure 4). Previous work showed that the cell ratio of alga and yeast in a similar synthetic ecological system stabilized at a specific ratio, even when the organisms were inoculated at a number of different inoculation dosages [25]. Therefore, the ideal yeast/bacteria cell ratio under these selective conditions could be further optimized.

### 3.4. Co-Evolution of S. cerevisiae and Lb. plantarum 

BY4742Δ*thi4* and IWBT B038 were grown under sequential co-culture conditions in the absence of lysine and isoleucine. After reaching approximately 100 generations, the co-cultures were plated on YPD and MRS agar, then three yeast and three bacteria colonies from each biological repeat were randomly selected. Each yeast colony was paired with the three bacteria colonies from the same pool (biological repeat) and the growth of the pairings was monitored for 8 days in SM medium in the absence of lysine and isoleucine. All the evolved pairings grew better than the unevolved parental pairing. The best performing pairing from each pool was selected for further screening (Appendix A).

For the second screening, the selected evolved isolates were paired with each other and also with the parental strains (Appendix A). In addition to the selective conditions (Figure 5a), all pairwise combinations were also grown under non-selective conditions, where no amino acids were omitted (Figure 5c). The isolates were also grown in monoculture to ensure that they were not able to grow in the absence of their partner when the selected amino acids were omitted, and also to see how they performed in monoculture when all the amino acids necessary for their growth were already present in the environment (Figure 5b). 

Interestingly, and perhaps not surprisingly, the poorest performing pairings under selective conditions (Figure 5a) were the evolved bacterial isolates (B312, B322, and B331) paired with the unevolved parental yeast strain BY4742Δ*thi4* (THI4). This indicates that the evolved yeast isolates are better adapted to support the growth of the bacteria. The evolved bacterial isolates also appear to derive benefits other than a source of isoleucine from the evolved yeast isolates, since the unevolved parental bacteria strain (B038) outperformed them in monoculture under non-selective conditions (Figure 5b). Furthermore, these evolved bacterial isolates also showed relatively poor growth when paired with the parental yeast (THI4) under non-selective conditions (Figure 5c). These results are significant and support the hypothesis that additional mutually supportive adaptations unrelated to the primary selection pressure will emerge in a co-evolving population where species are forced to depend on one another for growth. Similar improvements in the growth of co-evolved *S. cerevisiae* and *Chlorella sorokiniana* isolates were seen in a synthetic ecological system, where the two species rely on each other for nitrogen and carbon, respectively [26].

The best performing pairings under selective conditions (Figure 5a) were the evolved yeast isolates (Y311, Y323, and Y331) paired with the unevolved parental bacterial strain IWBT B038. Previous experiments showed that the yeast starts growing when the bacteria enters the stationary phase (Figure 4), indicating that the release of amino acids by the bacteria may be due to nutrients leaking into the environment following cell death or due to a more permeable bacterial membrane during the later growth stages. Ethanol and other wine components affect the membrane properties of LAB [27], while *Lb. plantarum* WCFS1 has been shown to modify the fatty acid composition of the cell membrane when exposed to ethanol [28]. Continuous exposure to the harsh fermentation conditions and increasing ethanol concentrations during co-evolution likely selected for bacterial isolates with better adapted membranes, and therefore also isolates that do not release amino acids as efficiently as their parental strain.

These initial experiments show interesting growth patterns between the parental and evolved isolates, indicating that genetic or transcriptomic changes occurred that are beneficial to the growth of the bacteria. It is difficult to speculate which changes may have occurred, due to the uncertainty and complexity surrounding the interactions between *S. cerevisiae* and *Lb. plantarum* under these specific conditions. However, previous interaction studies suggest that glucose metabolism and ethanol production in yeast [23] and lipid biosynthesis in LAB [24] may be worth investigating. The assumption can be made that increased glucose consumption and subsequent ethanol production by the evolved yeast isolates would create conditions selective for bacteria with higher ethanol tolerance. Additionally, large-scale chromosomal rearrangements have been shown to occur in *Lachancea kluyveri* following co-evolution under competitive conditions with different bacterial species [29]. Therefore, whole-genome sequencing of the evolved yeast isolates from this study will provide further insights by identifying genetic changes that occurred during the co-evolution of the species. Understanding how these mutations benefit the bacteria could also highlight some of the complex ways these strains interact with each other.

Previous studies have shown that evolution strategies are invaluable tools that can be used for the improvement of wine-related microorganisms, such as the use of directed evolution to increase ethanol tolerance in bacteria [19] and co-evolution for the selection of more diverse flavor and aroma profiles in yeast [30]. In this study, we developed a novel evolution system, which had not previously been reported to the best of our knowledge, where *S. cerevisiae* and *Lb. plantarum* are co-evolved under conditions where they are forced to depend on each other through the reciprocal exchange of amino acids. Evolving obligate cooperative interactions will result in the selection of isolates that better support the growth of their partner species, either by strengthening the initial interaction or by developing additional mutually beneficial traits. Here, we showed that after approximately 100 generations—a relatively short evolutionary time period—there was a significant improvement in the mutualistic growth of the evolved isolates and the emergence of additional yeast adaptations that are beneficial to the bacteria. This is, therefore, a powerful tool that can be used to better understand the complex interactions within microbial ecosystems and how these interactions drive the evolution of species. Furthermore, this system can also be used for the development of optimized yeast–bacteria pairings with more complementary nutrient requirements for use in winemaking. Further work should still be done to better understand how changes in the yeast and bacteria population sizes, which consequently influence the supply and demand for amino acids, affect the stability of the established mutualism, as this was shown to be an important consideration when developing these nutrient dependent systems. 

## Figures and Tables

**Figure 1 microorganisms-08-01109-f001:**
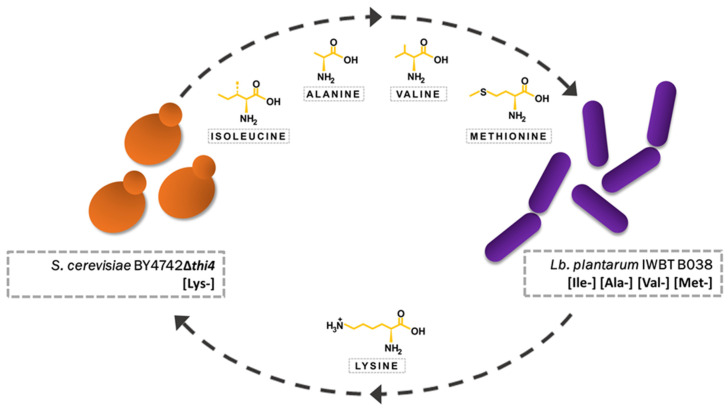
In the established synthetic mutualism *S. cerevisiae* strain BY4742Δ*thi4* and *Lb. plantarum* strain IWBT B038 provide their partner with the required amino acids omitted from their external environment.

**Figure 2 microorganisms-08-01109-f002:**
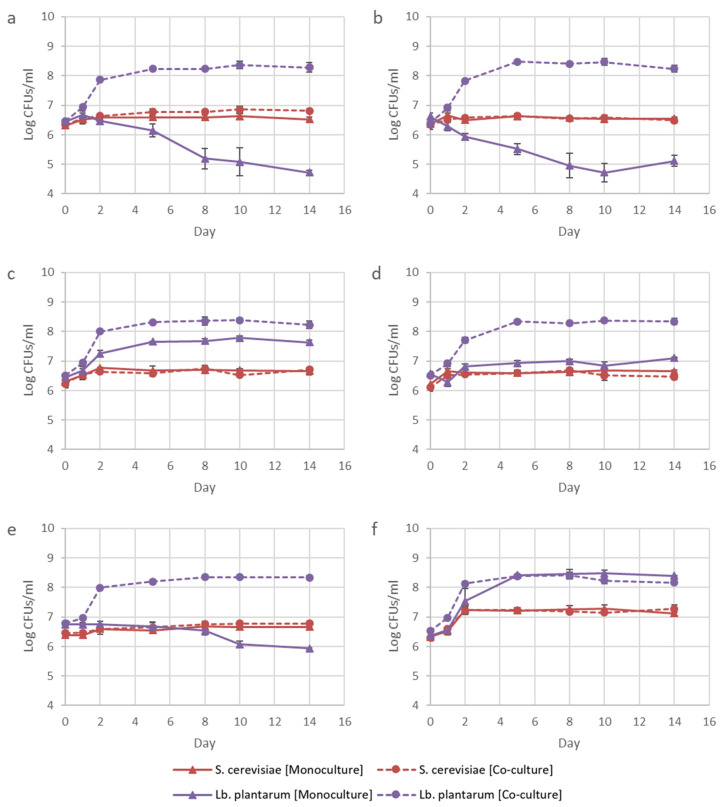
Log CFUs/mL for *S. cerevisiae* strain BY4742Δ*thi4* and *Lb. plantarum* strain IWBT B038 over the 14-day incubation period for the Lys-Ile (**a**), Lys-Ala (**b**), Lys-Val (**c**), Lys-Met (**d**), Lys-Ile-Ala-Val-Met (**e**), and control (**f**) treatments. Data shown are means of triplicates, with error bars representing standard deviation.

**Figure 3 microorganisms-08-01109-f003:**
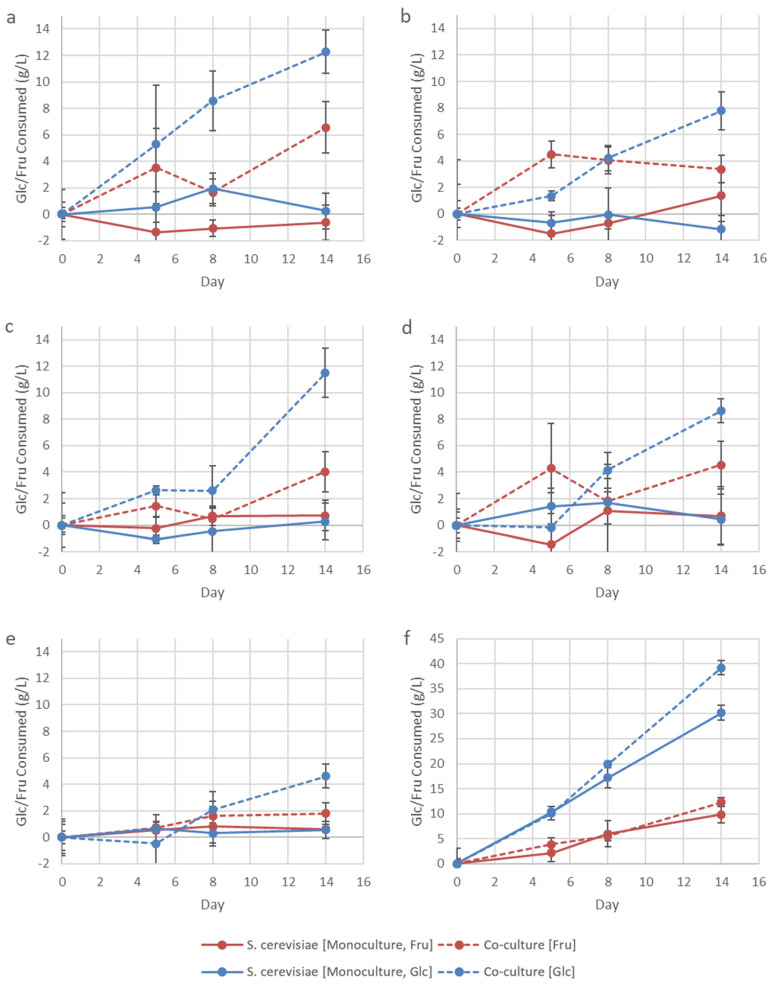
Total glucose (Glc) and fructose (Fru) consumed over the 14-day incubation period by *S. cerevisiae* mono- and co-cultures for the Lys-Ile (**a**), Lys-Ala (**b**), Lys-Val (**c**), Lys-Met (**d**), Lys-Ile-Ala-Val-Met (**e**), and control (**f**) treatments. Data shown are means of triplicates, with error bars representing standard deviation.

**Figure 4 microorganisms-08-01109-f004:**
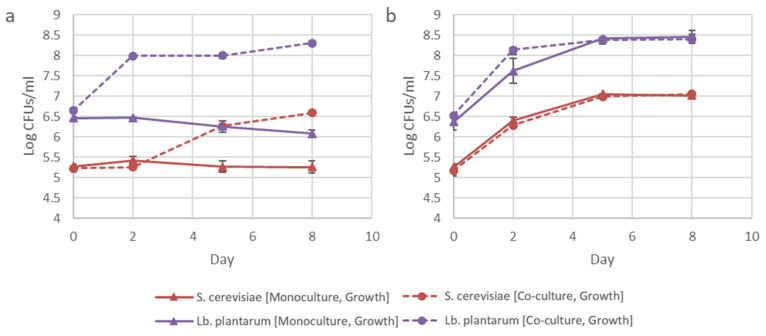
Influence of inoculation dosage on log CFUs/mL for *S. cerevisiae* BY4742Δ*thi4* and *Lb. plantarum* IWBT B038 under mono- and co-culture conditions for Lys-Ile (**a)** and control (**b**) treatments. Data shown are means of triplicates, with error bars representing standard deviation.

**Figure 5 microorganisms-08-01109-f005:**
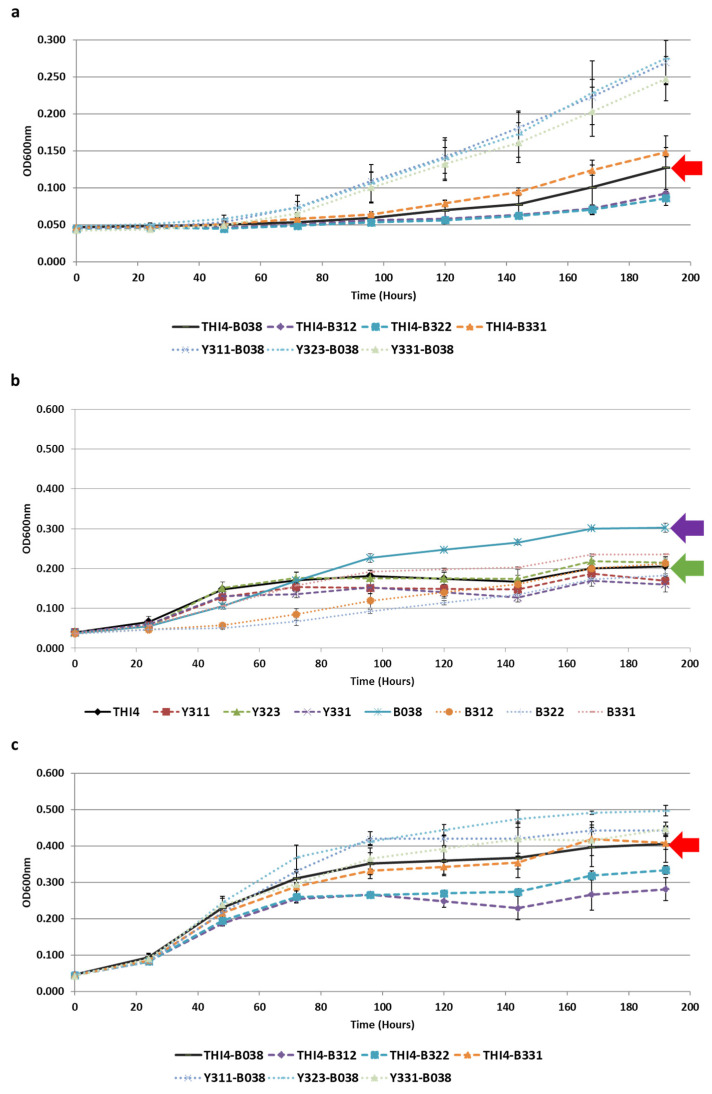
Growth of *S. cerevisiae* BY4742Δ*thi4* and *Lb. plantarum* IWBT B038 evolved isolates in the absence of lysine and isoleucine (**a**) and presence of all amino acids (**b**,**c**). (**a**,**c**) The paired growth of the isolates and (**b**) the growth of each isolate in monoculture. Red arrows show the growth of the unevolved parental pairing THI4-B038 (**a**,**c**) and the purple and green arrows show the growth of the parental bacterial and yeast strains, respectively (**b**). Data shown are means of triplicates, with error bars representing standard deviation.

**Table 1 microorganisms-08-01109-t001:** Chemical composition of the synthetic medium (SM), simulating standard grape juice, adjusted from previous descriptions [20,21].

Compound Group	Compound	Amount per Liter
**Carbon sources**	Glucose	100 g
	Fructose	100 g
**Acids**	Potassium L-Tartrate Monobasic	2.5 g
	L-Malic acid	4 g
	Citric acid	0.2 g
**Salts**	Potassium phosphate dibasic (K_2_HPO_4_)	1.14 g
	Magnesium sulphate heptahydrate (MgSO_4_.7H_2_O)	1.23 g
	Calcium chloride dihydrate (CaCl_2_.2H_2_O)	0.44 g
**Nitrogen sources**	Ammonium sulphate	0.3 g
	Amino acids (prepared as 100X stock solution in 20 g/L NaHCO_3_ buffer solution)	
	- Tyrosine	0.014 g
	- Tryptophane	0.137 g
	- Isoleucine	0.025 g
	- Aspartic acid	0.034 g
	- Glutamic acid	0.092 g
	- Arginine	0.286 g
	- Leucine	0.037 g
	- Threonine	0.058 g
	- Glycine	0.014 g
	- Glutamine	0.386 g
	- Alanine	0.111 g
	- Valine	0.034 g
	- Methionine	0.024 g
	- Phenylalanine	0.029 g
	- Serine	0.060 g
	- Histidine	0.025 g
	- Lysine	0.013 g
	- Cysteine	0.010 g
	- Proline	0.468 g
**Trace elements (prepared as 100X stock solution)**	Manganese(II) chloride tetrahydrate (MnCl_2_.4H_2_O)	200 µg
	Zinc(II) chloride (ZnCl)	135 µg
	Iron(II) chloride (FeCl_2_)	30 µg
	Copper(II) chloride (CuCl_2_)	15 µg
	Boric acid (H_3_BO_3_)	5 µg
	Cobalt(II) nitrate hexahydrate (Co(NO_3_)_2_.6H_2_O)	30 µg
	Sodium molybdate dihydrate (NaMoO_4_.2H_2_O)	25 µg
	Potassium iodate (KIO_3_)	10 µg
**Vitamins (prepared as 100X stock solution)**	Myo-inositol	100 mg
	Pyridoxine hydrochloride	2 mg
	Nicotinic acid	2 mg
	Calcium pantothenate	1 mg
	Thiamin hydrochloride	0.5 mg
	Para-aminobenzoic acid (PABA)-K	0.2 mg
	Riboflavin	0.2 mg
	Biotin	0.125 mg
	Folic acid	0.2 mg
**Anaerobic factors (prepared as 10X stock solution in hot 96% EtOH)**	Ergosterol	10 mg
	Tween 80	0.5 mL

**Table 2 microorganisms-08-01109-t002:** Amino acid treatments used in this study.

Treatment Name	Amino Acids Omitted	Yeast Assimilable Nitrogen (YAN) Concentration (mg/L)
Control	None	322.48
Lys-Ile	Lysine	317.32
	Isoleucine	
Lys-Ala	Lysine	302.54
	Alanine	
Lys-Val	Lysine	315.93
	Valine	
Lys-Met	Lysine	317.74
	Methionine	
Lys-Ile-Ala-Val-Met	Lysine	293.55
	Isoleucine	
	Alanine	
	Valine	
	Methionine	

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
