# Peer review of "Enforced Mutualism Leads to Improved Cooperative Behavior between Saccharomyces cerevisiae and Lactobacillus plantarum"

_microorganisms, 2020, doi:10.3390/microorganisms8081109_

Round 1

Reviewer 1 Report

In this paper, du Toit et al. asses the applicability of a synthetic echological approach to better control and optimise the interactions between Saccharomyces cerevisiae and Lactobacillus plantarum strains, auxotrophic for specific ammino acids. The paper is well written, methodological approach is correct, topic is of great concern. Statistical approach is fine, as well as figures and tables. Results are clear and well discussed. Overall, the paper is interesting and in line with a modern concept of wine microbiology.

Author Response

Thank you for the positive comments. No modifications were suggested.

Reviewer 2 Report

Paper: Saccharomyces cerevisiae and Lactobacillus plantarum cooperation in a synthetic wine environment described by du Toit et al.

Comments:

L 11  malolactic fermentation (AF and MLF), respectively.

In introduction Authors should focus much more and exposed some novelty this proposition

The aim – should be describe more clearly

Section 2.1 – add some information about city, country etc

Description: S. cerevisiae and Lb. plantarum – should be in all Figure as italic

Fig 5 – should be corrected, ist not possible to look some data, especially on part a and c

Additionally these data present on Fig. 5 statistically differences ?

Authors should add some more information about what happen in real wine about bacteria and yeast, they cooperation or not ? Any information in literature part ?

Author Response

L 11  malolactic fermentation (AF and MLF), respectively.

Corrected in manuscript: Page 1 Line 14

In introduction Authors should focus much more and exposed some novelty this proposition

Indeed, we agree that we did not highlight sufficiently the novelty of the approach.

To address this issue, we have changed the title and made changes to the abstract. In the introduction, Lines 75 – 80 were rewritten as follows:  

“Furthermore, directed evolution has been suggested as a strategy to further improve strain compatibility and to optimize strains by increasing their resistance to inhibitory biotic and abiotic factors encountered under winemaking conditions [18,19]. The chemical and biological complexity of grape must make it difficult to study the interactions between specific strains and to monitor the evolution of these interactions over time. The use of simplified ecological systems is therefore necessary.”

The aim – should be describe more clearly

Clarified in manuscript: Page 2 Line 84 – 91

Here we assess the applicability of a novel synthetic ecology and coevolution approach to better understand and optimise the interactions between S. cerevisiae and Lb. plantarum strains. Nutrient co-dependency between the strains was used to establish obligatory mutualistic conditions, with the aim to evaluate if mutually beneficial adaptations independent of the obligatory nutrient exchange would emerge within the population over time. The advantage of the system is that the co-dependency ensures persistence and co-existence of both species over time, and should focus selection pressure to favor improved mutualistic traits.

Section 2.1 – add some information about city, country etc

Clarified in manuscript: Page 3 Line 103

“Strain IWBT B038 was previously isolated from grape must at SAGWRI.”

Stellenbosch (city) and South Africa (country) is already mentioned in the text [Line 102 – 103]

Description: S. cerevisiae and Lb. plantarum – should be in all Figure as italic

S. cerevisiae and Lb. plantarum are italicized in all figure headings. Unfortunately, in Excel it is not possible to format individual characters within the figure legends.

Fig 5 – should be corrected, ist not possible to look some data, especially on part a and c

Corrected in manuscript: Page 12 Line 444 – 448

Figure 5: Frame a and c from original manuscript was moved to supplementary material (as Figure S2). The modified Figure 5: Frame a and c in the current manuscript only shows the paired growth of the evolved isolates with the parental strains.

Additionally these data present on Fig. 5 statistically differences ?

Corrected in manuscript: Page 12 Line 444 – 448

Error bars in Figure 5 are now more visible

Authors should add some more information about what happen in real wine about bacteria and yeast, they cooperation or not ? Any information in literature part ?

This type of obligatory mutualism, centered on the reciprocal exchange of amino acids, would not occur in grape must where amino acids and proteins are naturally present. The interactions between yeast and bacteria in grape must/wine are mentioned in lines 48 – 73.

Reviewer 3 Report

The manuscript describes a study to assess applicability of a synthetic ecological approach to better control and optimise the interactions between S. cerevisiae and Lb. plantarum strains, auxotrophic for specific amino acids. In the future, the results of this study could be of great use to wine makers.

Suggestions:

In this regard, the following are suggestions to keep in mind to improve the quality and the understanding of the manuscript text.

1.- Title

Authors should choose a title that reflects better the content and the aim of the manuscript. The term “synthetic wine” is not very correct from an oenological point of view.

2.-  Abstract

Authors should rewrite the abstract and they should review the style of structured abstract according to the indication of the journal.

The abstract should be a single paragraph and should follow the style of structured abstracts, but without headings: 1) Background: Place the question addressed in a broad context and highlight the purpose of the study; 2) Methods: Describe briefly the main methods or treatments applied. Include any relevant preregistration numbers, and species and strains of any animals used. 3) Results: Summarize the article's main findings; and 4) Conclusion: Indicate the main conclusions or interpretations. The abstract should be an objective representation of the article: it must not contain results which are not presented and substantiated in the main text and should not exaggerate the main conclusions.

3.- Introduction

Lines 73-82: Authors should rewrite the objectives of the paper. This is necessary to understand the purpose of the manuscript

4.- Materials and Methods

Lines 105 and 110: Replace “Synthetic grape juice-like (SGJ)” with “synthetic medium (SM) that simulates standard grape juice”.

Rewrite the acronym SGJ with the new nomenclature

5.- Results and Discussion

Figure 1 should go as supplementary. This is additional information.

Authors should conduct a review to incorporate more up-to-date bibliographic citations.

Author Response

Thank you for the detailed comments

Reviewer 3

1.- Title

Authors should choose a title that reflects better the content and the aim of the manuscript. The term “synthetic wine” is not very correct from an oenological point of view.

Changed title of manuscript: Page 1 Line 2

Enforced mutualism leads to improved cooperative behaviour between Saccharomyces cerevisiae and Lactobacillus plantarum

2.-  Abstract

Authors should rewrite the abstract and they should review the style of structured abstract according to the indication of the journal.

The abstract should be a single paragraph and should follow the style of structured abstracts, but without headings: 1) Background: Place the question addressed in a broad context and highlight the purpose of the study; 2) Methods: Describe briefly the main methods or treatments applied. Include any relevant preregistration numbers, and species and strains of any animals used. 3) Results: Summarize the article's main findings; and 4) Conclusion: Indicate the main conclusions or interpretations. The abstract should be an objective representation of the article: it must not contain results which are not presented and substantiated in the main text and should not exaggerate the main conclusions.

The abstract may indeed have been misleading about the focus of the work. We hope that the new abstract captures the essence of our work more appropriately (Page 1 Line 13):

“Saccharomyces cerevisiae and Lactobacillus plantarum are responsible for alcoholic and malolactic fermentation (AF and MLF), respectively. Successful completion of both fermentations is essential for many styles of wine, and an understanding of how these species interact with each other, as well as the development of compatible pairings of these species, will help to manage the process. However, targeted improvements of species interactions are difficult to perform, in part because of the chemical and biological complexity of natural grape juice. Synthetic ecological systems reduce this complexity and can overcome these difficulties. In such synthetic systems, mutualistic growth of different species can be enforced through the reciprocal exchange of essential nutrients. Here, we implemented a novel approach to evolve mutualistic traits by establishing a co-dependent relationship between S. cerevisiae BY4742Δthi4 and Lb. plantarum IWBT B038 by omitting different combinations of amino acids from a chemically defined synthetic grape juice-like media. After optimization, the two species were able to support the growth of each other when grown in the absence of appropriate combinations of amino acids. In these obligatory mutualistic conditions, BY4742Δthi4 and IWBT B038 were coevolved for approximately 100 generations. Selected evolved isolates showed improved mutualistic growth and the growth patterns under non-selective conditions indicate the emergence of mutually beneficial adaptations independent of the synthetic selection pressure. The combined use of synthetic ecology and coevolution is a promising strategy to better understand and biotechnologically improve microbial interactions.”

3.- Introduction

Lines 73-82: Authors should rewrite the objectives of the paper. This is necessary to understand the purpose of the manuscript

Clarified in manuscript: Page 2 Line 75 – 91

Furthermore, directed evolution has been suggested as a strategy to further improve strain compatibility and to optimize strains by increasing their resistance to inhibitory biotic and abiotic factors encountered under winemaking conditions [18,19]. The chemical and biological complexity of grape must make it difficult to study the interactions between specific strains and to monitor the evolution of these interactions over time. The use of simplified ecological systems is therefore necessary.

Here we assess the applicability of a novel synthetic ecology and coevolution approach to better understand and optimise the interactions between S. cerevisiae and Lb. plantarum strains. Nutrient co-dependency between the strains was used to establish obligatory mutualistic conditions, with the aim to evaluate if mutually beneficial adaptations independent of the obligatory nutrient exchange would emerge within the population over time. The advantage of the system is that the co-dependency ensures persistence and co-existence of both species over time, and should focus selection pressure to favour improved mutualistic traits.

4.- Materials and Methods

Lines 105 and 110: Replace “Synthetic grape juice-like (SGJ)” with “synthetic medium (SM) that simulates standard grape juice”.

Rewrite the acronym SGJ with the new nomenclature

Corrected in manuscript: Page 3 Line 122 and 128

Acronym SGJ replaced with SM in manuscript

5.- Results and Discussion

Figure 1 should go as supplementary. This is additional information.

Authors should conduct a review to incorporate more up-to-date bibliographic citations.

Figure 1 is a clear visual representation of our established synthetic mutualism which makes the concept easier to understand for readers who are unfamiliar with the field. Therefore, it was decided to leave the figure in the main text.

Included 2019 review by Bartle et al.